# Computer vision detects covert voluntary facial movements in unresponsive brain injury patients

Xi Cheng[1,2], Sujith Swarna[1], Jermaine Robertson[1], Nathaniel A. Cleri[1,3], Jordan R. Saadon[1,4], Chiemeka Uwakwe[1], Yindong Hua [1,2], Seyed Morsal Mosallami Aghili [1], Cassie Wang[1,5], Robert S. Kleyner[1], Xuwen Zheng[1], Ariana Forohar[1], John Servider[1], Kurt Butler[2], Chao Chen[6], Jordane Dimidschstein[7], Petar M. Djurić[2], Charles B. Mikell [1] ✉ & Sima Mofakham [1,2] ✉

## Abstract

**Background** Many brain injury patients who appear unresponsive retain subtle, purposeful motor behaviors, signaling capacity for recovery. We hypothesized that low-amplitude movements precede larger-amplitude voluntary movements detectable by clinicians after acute brain injury. To test this hypothesis, we developed a novel, as far as we are aware, computer vision-based tool (SeeMe) that detects and quantifies low-amplitude facial movements in response to auditory commands.

**Methods** We enrolled 16 healthy volunteers and 37 comatose acute brain injury patients (Glasgow Coma Scale ≤8) aged 18–85 with no prior neurological diagnoses. We measured facial movements to command assessed using SeeMe and compared them to clinicians' exams. The primary outcome was the detection of facial movement in response to auditory commands. To assess comprehension, we tested whether movements were specific to command type (i.e., eye-opening to open your eyes and not stick out your tongue) with a machine learning-based classifier.

**Results** Here we show that SeeMe detects eye-opening in comatose patients 4.1 days earlier than clinicians. SeeMe also detects eye-opening in more comatose patients (30/36, 85.7%) than clinical examination (25/36, 71.4%). In patients without an obscuring endotracheal tube, SeeMe detects mouth movements in 16/17 (94.1%) patients. The amplitude and number of SeeMe-detected responses correlate with clinical outcome at discharge. Using our classifier, eye-opening is specific (81%) to the command open your eyes.

**Conclusion** Acute brain injury patients have low-amplitude movements before overt movements. Thus, many covertly conscious patients may have motor behavior currently undetected by clinicians.

## Plain language summary

Some people with severe brain injuries may appear unconscious, but still have some awareness and the ability to move. These movements are often too small to be seen by doctors during routine exams. In this study, researchers created a computer vision tool called SeeMe that can detect tiny facial movements in response to voice commands. They tested this tool on both healthy people and patients in a coma. SeeMe found signs of movement several days before doctors could. It also found more patients who showed responses than clinical exams did. This suggests that some patients may be aware earlier than expected. Tools like SeeMe could help doctors identify hidden signs of recovery and make better decisions about patient care.

Assessing the level of consciousness in the acute brain injury (ABI) population is challenging. In the clinical setting, consciousness is usually assessed by asking the patient to follow auditory commands (for example, squeeze my hand or wiggle your toes). Command following confirms that the patient is capable of understanding the verbal content of the command and moving in response. However, many patients appear to be aware without following commands; 15–25% of ABI patients suffer from cognitive-motor dissociation or covert consciousness, i.e., they are aware but show no overt signs of consciousness[1,2]. Investigators have used various techniques to identify preserved consciousness in unresponsive patients, but none have yet come

[1]Department of Neurosurgery, Renaissance School of Medicine at Stony Brook University, Stony Brook, NY, USA. [2]Department of Electrical and Computer Engineering, Stony Brook University, Stony Brook, NY, USA. [3]Department of Surgery, Donald and Barbara Zucker School of Medicine at Hofstra/Northwell, Manhasset, NY, USA. [4]Department of Neurosurgery, University of Maryland Medical Center, Baltimore, MD, USA. [5]Department of Internal Medicine, Donald and Barbara Zucker School of Medicine at Hofstra/Northwell, Manhasset, NY, USA. [6]Department of Biomedical Informatics, Stony Brook University, Stony Brook, NY, USA. [7]Stanley Center for Psychiatric Research, Broad Institute of Harvard and MIT, Cambridge, MA, USA. ✉e-mail: charles.mikell@stonybrookmedicine.edu; sima.mofakham@stonybrookmedicine.edu

into widespread use. Electroencephalography (EEG) evidence of response to complex linguistic input was recently shown to correlate with neurological outcomes after brain injury, even when there were no overt motor responses[3,4]. Functional magnetic resonance imaging (fMRI) responses to commands have also shown evidence of conscious processing even when overt motor behavior is not seen[5,6]. However, failure to overtly follow commands can reflect dysfunction at any step between auditory input and motor output[7]. This gap in assessment has motivated techniques such as the Motor Behavior Tool[8,9], which is designed to identify conscious behavior that may be missed by other rating systems, such as the Coma Recovery Scale-Revised (CRS-R)[10]. Still, the Motor Behavior Tool and all other scoring systems depend on subjective evaluation and identification of non-reflex behavior. These non-reflex behaviors may be very low-amplitude and difficult for clinicians to detect[7].

Therefore, we hypothesized that patients returning to consciousness would develop low-amplitude movements before larger-amplitude movements identifiable by clinicians. To test this hypothesis, we developed a computer vision-based technique we call SeeMe to identify low-amplitude facial movements in response to verbal stimuli. The face has a large, bilateral representation in the cortex[11] and innumerable small muscular attachments[12], leading us to predict that early signs of neurological recovery would be manifested in the face. Using our tool, we assess whether patients recovering consciousness move their faces in response to auditory commands. We find that SeeMe detects facial movements earlier and in more patients than clinical examination, and that the amplitude and frequency of these movements correlate with functional outcomes at discharge.

## Methods

### Study population
This prospective cohort study was conducted at Stony Brook University Hospital from June 25, 2021, to August 27, 2024. Adult comatose patients (aged 18–85 years) were consecutively enrolled following ABI (traumatic brain injury (TBI), spontaneous subarachnoid hemorrhage, severe meningoencephalitis, etc.). Comatose patients were defined as those with an initial Glasgow Coma Score (GCS) ≤8. Healthy subjects were also enrolled for model creation and comparison (Table 1). A history of a neurologically debilitating disease (i.e., neoplasia, Alzheimer's, multiple sclerosis, major vessel stroke, previous severe TBI, etc.) excluded participants from enrollment.

### Standard protocol approvals, registrations, and patient consents
Study approval was obtained from the Stony Brook University Institutional Review Board (IRB2019-00199). The study complied with all relevant regulations. Written informed consent was obtained from healthy subjects and the ABI patients' legally authorized representatives. The faces of patients with ABIs were blurred per IRB approval. Example frames are from healthy subjects who consented to show their faces in publication. This study is registered on ClinicalTrials.gov (SeeMe: an Automated Tool to Detect Early Recovery After Brain Injury, NCT06083441, Enrollment June 2019–June 2025).

### Data collection
ABI patients were examined by the study team daily on weekdays, except when clinical status prevented safe examination. Medications such as sedatives and muscle paralytics were given based on the decision of the patient's clinical team, who also determined patient management. When medically safe to do so, these medications were paused at least 15–30 min before study sessions. The level of sedation was documented during each session (Supplementary Table 1). Each study visit began with a video-recorded CRS-R score assessment by a trained member of the research team (trained medical students)[10]. Single-use headphones were then placed in the patient's ears for the Auditory Stimulation study portion.

### SeeMe algorithm
SeeMe quantifies facial movements in response to spoken commands in ABI patients by vector field analysis. The algorithm tags individual facial pores

(resolution ~0.2 mm) and tracks their movements in response to the presented stimulus[13]. SeeMe presented each subject with three commands: (1) Stick out your tongue, (2) Open your eyes, and (3) Show me a smile using the PsychoPy software (an open-source Python-based software for running experiments) while videotaping patients' responses[14]. These commands were chosen to target various facial regions and musculature. Each command video began with a 1-min resting baseline recording of the subject's face with no command presentation. The command was then presented in blocks of ten, with 30–45 s (±1 s jitter) between commands. We outline the data collection and video processing pipeline (Fig. 1).

### Clinical and blinded raters
To evaluate SeeMe against the current clinical examinations (CRS-R and GCS) and the naked eye, two independent blinded raters (trained medical students) were assigned to watch each video and assess the existence of stimulus-evoked movement in response to the presented auditory commands. These blinded raters consisted of study team members without knowledge of SeeMe's computed results and the patient's clinical examination scores. Blinded raters were instructed to watch the recorded videos of the patient's face while commands were presented and to rate the patient's response to each command as either responsive or unresponsive in a yes or no fashion. The criteria used by the blinded raters to determine a positive response were as follows: The movement had to arrive within 20 s of the command presentation. The movement had to come in the corresponding region of interest (ROI) (e.g., Stick out your tongue had to elicit a movement in the mouth area to be considered valid). The movement had to be a deviation from the patient's baseline. A patient who was constantly blinking was not considered responding to the Open your eyes command unless there was a new movement, such as opening their eyes wider, after the command presentation. The movement must not have been caused by an artifact (e.g., when a doctor or clinician enters the field of view, bed/camera movement, or a change in room lighting).

This resulted in 10 ratings per video per blinded rater (20 in total). A command was considered positive by the blinded raters if either of the blinded raters thought there was a response. If a video scored three positive ratings (out of ten), then the patient was said to be responsive in that video. The blinded raters were also tasked with determining which videos contained artifacts. After all videos had been assessed, the data were compared to SeeMe's results. 82.5% of the ratings were consistent between the two blinded raters, and Cohen's kappa was 0.44. CRS-R scores were calculated for each patient before the auditory commands were played. A CRS-R auditory score > 2, defined as reproducible and/or consistent movement in response to command, was considered a sign of consciousness and was compared to the SeeMe mouth command video results. GCS scores were collected regularly by a clinical examiner (the patient's clinical care team). A GCS eye subscore (GCS-E) > 2, defined as eye-opening in response to speech or spontaneously, was considered a sign of consciousness and was compared to the SeeMe eye command video results. GCS was used for this analysis because it differentiates eye-opening to voice (GCS-E = 3) from eye-opening to painful stimulation (GCS-E = 2). By contrast, CRS-R only identifies a single type of stimulus-evoked eye-opening (CRS-R-arousal = 1, eye-opening to stimulation).

### Outcome measures
The primary outcome measure investigated was time from ABI to detect stimulus-evoked facial movement, assessed using SeeMe, clinical examiners, and blinded raters. Secondary measures included the GCS at discharge, the Glasgow Outcome Scale-Extended (GOS-E) at discharge and 6 months/most recent, and whether the patient was determined to be following commands at discharge. ABI patients were examined daily by their clinical care team for command following ability. During that examination, the

**Table 1 | Study population demographics, clinical information, and in-hospital complications**

| | N<br>Median age (IQR), years | Total population<br>53<br>44 (36) | Healthy cohort<br>16<br>27 (2) | Comatose cohort<br>37<br>57 (27) |
|---|---|---|---|---|
| Sex | Male | 40 (75.5%) | 11 (68.8%) | 29 (78.4%) |
| | Female | 13 (24.5%) | 5 (31.3%) | 8 (21.6%) |
| Race | White | 41 (77.3%) | 13 (81.3%) | 28 (75.7%) |
| | Black or African American | 2 (3.8%) | 0 (0.0%) | 2 (5.4%) |
| | Asian | 4 (7.5%) | 3 (18.8%) | 1 (2.7%) |
| | American Indian or Alaskan native | 1 (1.9%) | 0 (0.0%) | 1 (2.7%) |
| | Other/Unknown | 5 (9.4%) | 0 (0.0%) | 5 (13.5%) |
| Ethnicity | Hispanic or Latino | 7 (13.2%) | 2 (12.5%) | 5 (13.5%) |
| | Non-Hispanic or Latino | 41 (77.3%) | 14 (87.5%) | 27 (73.0%) |
| | Other/Unknown | 5 (9.4%) | 0 (0.0%) | 5 (13.5%) |
| Clinical presentation | Mean initial CRS-R/Days from acute brain injury (SD) | | | 5.3 (5.2)/6.4 (4.7) |
| | Mean final CRS-R/Days from acute brain injury (SD) | | | 11.5 (6.7)/22.2 (16.6) |
| Types of acute brain injury | Traumatic brain injury | | | 27 (73.0%) |
| | Spontaneous intracerebral hemorrhage | | | 7 (18.9%) |
| | Meningoencephalitis | | | 1 (2.7%) |
| | Aneurysmal subarachnoid hemorrhage | | | 1 (2.7%) |
| | Hydrocephalus | | | 1 (2.7%) |
| Complications | Hospital-acquired pneumonia | | | 17 (45.9%) |
| | Delirium | | | 16 (43.2%) |
| | In-hospital hydrocephalus | | | 16 (43.2%) |
| | Cerebral edema | | | 15 (40.5%) |
| | Pulmonary embolism | | | 6 (16.2%) |
| | Aspiration pneumonia | | | 5 (13.5%) |
| | Convulsions | | | 4 (10.8%) |
| | In-hospital intracerebral hemorrhage | | | 4 (10.8%) |
| | Stroke | | | 4 (10.8%) |
| | Acute respiratory distress syndrome | | | 2 (5.4%) |
| | Acute renal failure | | | 2 (5.4%) |
| | Cerebrospinal fluid leak | | | 2 (5.4%) |
| | Sepsis | | | 2 (5.4%) |
| | Neurological infections | | | 1 (2.7%) |
| | Cardiac arrest | | | 0 (0%) |
| | Myocardial infarction | | | 0 (0%) |
| | Post-operative hemorrhagic shock | | | 0 (0%) |

clinical care team documented whether the patient followed commands and their GCS scores in the electronic medical record (EMR). GOS-E scores at discharge and 6 months/most recent were determined by independent raters blinded to each patient's hospital course and SeeMe results using EMR documentation.

## Statistics and reproducibility

Each auditory stimulation video was loaded into a Python-based video processing pipeline where machine learning techniques were used to estimate the patient's response to each command. Two variables quantified the response: Kolmogorov–Smirnov (KS) statistics and post-peak value. KS statistics reflect the magnitude of the patient's facial movement changes after command presentation, and the post-peak value demonstrates the magnitude of the overall facial displacement. For a deeper overview of this process, see Supplementary Methods.

After confirming that the data distribution was non-normal using the one-sample KS test, we utilized the Kruskal-Wallis test followed by post-hoc

pairwise comparisons using the Dunn-Bonferroni approach to analyze the correlation of SeeMe's performance with patients' outcomes. A chi-square test was also employed to analyze the association between categorical variables. All of these analyses were performed using IBM-SPSS Statistics V29.

## Deep neural network classifier

To ensure that stimulus-evoked responses were voluntary behaviors, not spontaneous movement or noise, we developed a deep neural network classifier to determine the command the patient responds to using the displacement of all the landmarks in each frame. We initially randomly split the data (80% into training and 20% into testing). Since the number of samples from the Open your eyes command was larger than that of the other two commands, we used data augmentation techniques to balance the sample number from different command types, thus avoiding sampling bias in the training data. We used a bidirectional long short-term memory (BiLSTM) network with a fully connected layer and a softmax layer for classification[15].

**Fig. 1 | The study design and data processing pipeline for SeeMe. a** Subject selection, data collection, and processing pipelines. **b** Experimental setup. To minimize vibrations, the camera is placed at the end of the hospital bed without touching any other equipment. It zooms in and focuses on the subject's face to capture the videos. Disposable headphones ensure that auditory stimulation commands are audible to the subject.

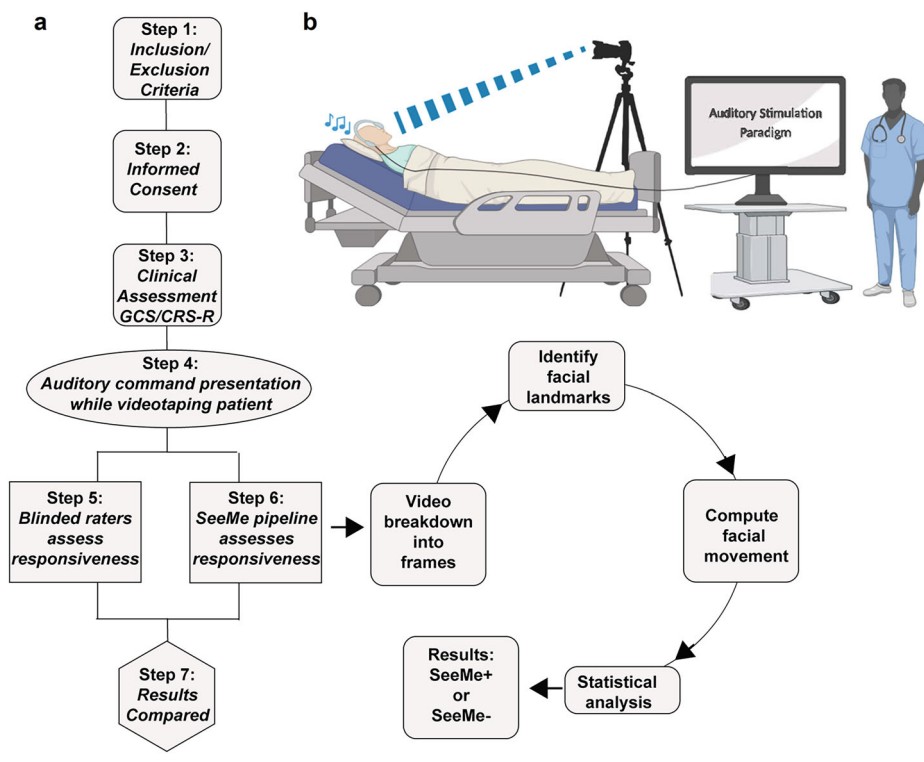

## Sedation control and complications

Whenever clinically feasible, sedation was paused 15–30 min before each SeeMe session. For all patients on continuous sedation infusion, we categorized the level of sedation as light and deep based on medication type and dosage. We categorized the infusion rate of <50 µg/kg/min of propofol[16–18], 0.2–0.7 µg/kg/min of dexmedetomidine (Precedex)[19], and 0.02–0.1 mg/kg/h of midazolam[20] as light sedation, and infusion rates exceeding these thresholds as deep sedation. Fentanyl, which is commonly administered to patients with ABI, shows sedative effects only at high doses. Thus, we considered fentanyl dosage lower than 1.5 mcg/kg/h IV for pain control as light sedation[21]. Due to the synergistic effects of Precedex and propofol, patients were classified as deep sedation with a propofol maintenance infusion threshold of 35 µg/kg/min when at least 0.5 µg/kg/h of Precedex was administered[22]. We also reported the most common complications with potential influence on the level of consciousness that overlapped with SeeMe sessions within one week of incidence[23]. Patients were then classified according to their sedation status/presence of concurrent complications and SeeMe video results (Supplementary Table 1).

## Reporting summary

Further information on research design is available in the Nature Portfolio Reporting Summary linked to this article.

## Results

### SeeMe detects low-amplitude stimulus-evoked movement hidden to the naked eye in ABI patients

We enrolled 37 ABI patients and 16 healthy subjects. In the ABI patients, we recorded 872 auditory stimulation videos (8565 single commands presented). Out of 872 auditory stimulation videos across 37 patients, we identified 423 analyzable videos (Supplementary Methods, 92 videos for Stick out your tongue, 246 for Open your eyes, and 85 for Show me a smile).

We had 36 patients with analyzable videos for the eye command and 17 with analyzable videos for the mouth commands. In the ABI patient cohort, SeeMe detected responses to the eye command (Open your eyes) in 30/36 patients and the mouth commands (Show me a smile and Stick out your tongue) in 16/17 patients (Supplementary Table 2). An example of single-trial heatmaps of facial movements for three patients is shown (Fig. 2a). The

time series of facial movements within the pre-specified ROI is associated with the three commands (i.e., the eyes for open your eyes) (Fig. 2b). Orange stars indicate movements confirmed as positive by SeeMe (KS statistic < 0.1 and post-peak amplitude > 400 [Supplementary Methods]). Videos were considered SeeMe+ if they contained at least 3/10 trials with stimulus-evoked movement (i.e., SeeMe+ trials). Otherwise, the videos were considered SeeMe-. The analysis of the healthy control cohort by SeeMe is also provided (Supplementary Fig. 1).

Two blinded raters reviewed the videos for stimulus-evoked movements for the ABI cohort. Blinded raters identified, on average, 2.8 ± 3.5 trials as responsive to the presented command, whereas SeeMe, on average, identified 5.4 ± 3.7 trials as responsive per session (out of 10 trials per command). SeeMe was much more sensitive to low-amplitude movements than blinded raters reviewing the videos (see Supplementary Results for a direct comparison and additional validation steps, Supplementary Fig. 2).

## Example cases

Subject 13 was an older adult involved in a motor vehicle collision. Initial computed tomography of the head (CTH) showed subarachnoid hemorrhage involving the left sylvian fissure and adjacent sulci, epidural hemorrhage around the right anterior temporal lobe, numerous skull and facial fractures, and pneumocephalus. Other injuries on presentation were right hemopneumothorax, rib fractures, dissection of the descending thoracic aorta, and left acetabular fractures. He was deeply comatose on admission (GCS 3T) and gradually improved. He opened his eyes on day 12, was detected by both SeeMe and GCS (GCS-E > 2), and followed motor commands on day 37. SeeMe detected stimulus-evoked mouth movement on day 18. His CRS-R improved to 8 by the end of the study. At 6-month follow-up, he lived with moderate to severe disability (GOS-E 3). For his stimulus-evoked eye movements, see Supplementary Video 1.

Subject 2 was a middle-aged individual who survived a bicycle–motor vehicle collision. Initial CTH showed left subdural hematoma along the frontal and parietal convexity with effacement of adjacent sulci; the remaining workup showed bilateral lower extremity injuries. He was also deeply comatose on admission (GCS 3T) and made a gradual recovery. He opened his eyes on day 22 (GCS-E > 2) and followed motor commands on day 48. SeeMe detected stimulus-evoked eye and mouth movements on day

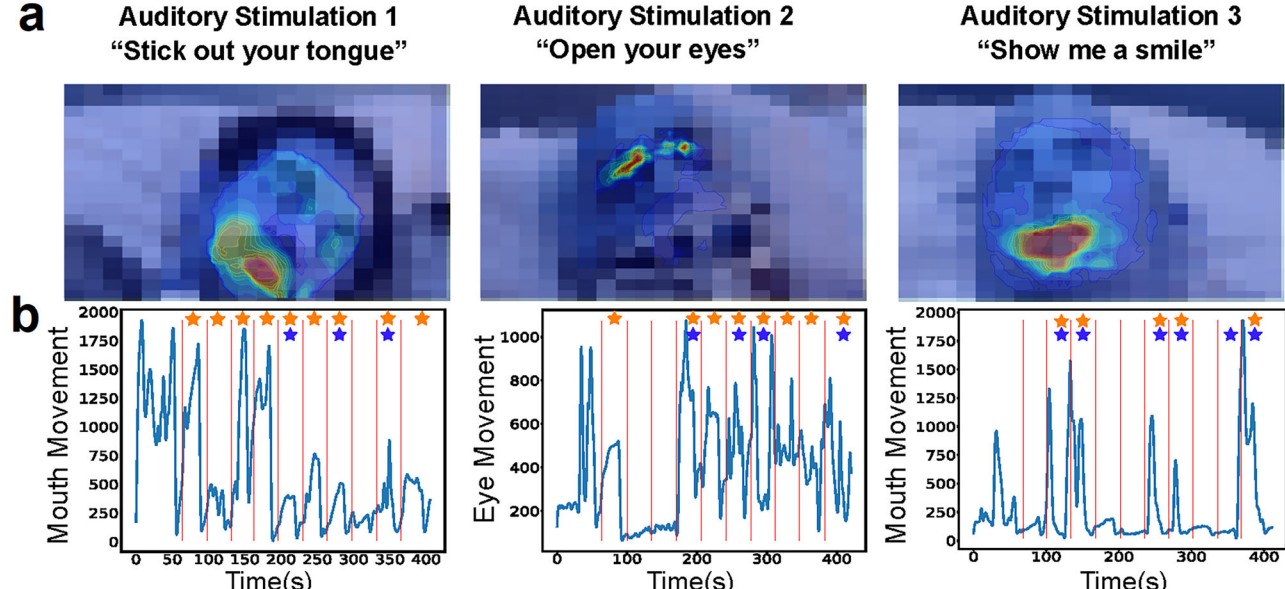

**Fig. 2 | SeeMe detects low-amplitude stimulus-evoked movements that precede the detection of consciousness by blinded raters in ABI comatose patients.** **a** Example heatmap results of the three auditory stimulation commands presented to comatose ABI patients. **b** The landmark summation plots for video trials associated with the frames above in **a** show the total movement in each auditory stimulation command's ROI. Blue lines show the movement summation as the trial progressed. The red vertical lines indicate the start of each auditory stimulation command. The orange stars indicate trials with positive responses as determined by SeeMe. Blue stars indicate positive trials as determined by our blinded raters (i.e., at least one blinded rater detected a response using the naked eye).

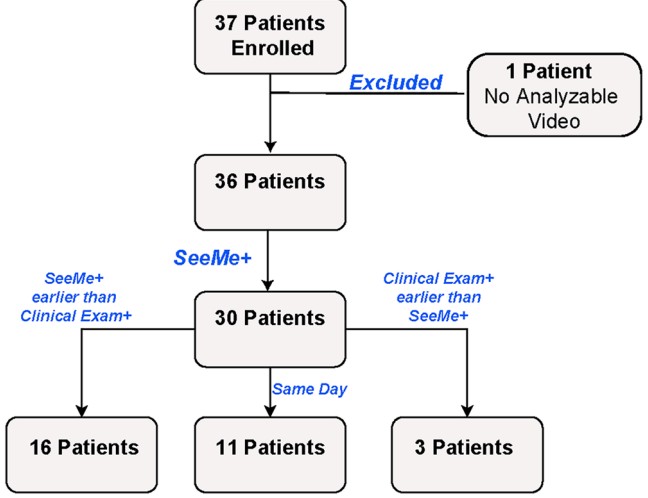

**Fig. 3 | SeeMe and clinical examination detection of recovery in the comatose ABI patient cohort.** This flowchart compares SeeMe's performance in detecting stimulus-evoked movements (SeeMe+) to clinical examination in ABI patients and patients' outcomes at discharge (GOS-E ≥ 3 were considered conscious outcomes).

19. His CRS improved to 16 by the end of the study. At 6-month follow-up, he lived with moderate to severe disability (GOS-E 3). For his mouth movements, see Supplementary Video 2.

### Low-amplitude stimulus-evoked movements precede clinical detection of consciousness in ABI patients

SeeMe detected movement (either eye or mouth) in 30/36 patients. In these patients, SeeMe detected movement before clinicians in 16/30 patients, on the same day in 11/30 patients, and after clinicians in 3/30 patients (Fig. 3).

SeeMe detected eye-opening on day $9.1 \pm 5.5$ (mean ± SD) after injury, while clinical examination detected eye-opening on day $13.2 \pm 11.4$ (mean ± SD). SeeMe's performance was compared against validated clinical examinations: the CRS-R (for mouth movements)[10]

and GCS (for eye-opening) (Fig. 4)[24]. GCS was used instead of CRS-R because of its granularity for eye-opening (differentiating eye-opening to voice from eye-opening to stimulation). For these patients, the average day of eye-opening detection via SeeMe (star) and clinical examination (circle) (GCS-E > 2) is depicted, marked on their average daily level of consciousness, as measured by GCS. Using the GCS-E, SeeMe detects eye-opening 4.1 days before clinical examiners.

We repeated this analysis with the mouth commands. In total, 16 patients had detectable, reproducible mouth movements via SeeMe. Excluding study dropouts and patients lacking video before auditory command following, seven also displayed command following measured by CRS-R (CRS-R auditory subscore > 2, reproducible movement to command). Five of these seven patients had mouth movements on SeeMe before clinicians, but two had clinical command-following before SeeMe-identified mouth movements (in both cases, there were problems with study procedures; see Supplementary Results). In these seven patients, SeeMe detected stimulus-evoked mouth movements on day $22.1 \pm 12.3$ (mean ± SD), while clinical examination detected command following movements on $30.4 \pm 20.4$. Their average daily CRS-R curve is shown, depicting the average day of mouth response via SeeMe (star) and CRS-R (circle) (Fig. 4b). Thus, SeeMe detected stimulus-evoked mouth movements 8.3 days earlier than clinical examiners.

### SeeMe's performance correlates with patients' outcomes

As our cohort of ABI patients recovered, the amplitude of their movements increased. The average eye and mouth movement amplitude on the first and last days that SeeMe had analyzable videos are provided (Supplementary Fig. 3).

We investigated the relationship between SeeMe performance and patients' functional outcomes at discharge as measured by GOS-E. SeeMe performance was assessed using both SeeMe+ trial response amplitude and the response detection frequency (fraction of SeeMe+ trials to the total number of trials). Both amplitude ($N = 293$, Kruskal–Wallis $w = 16.84$, $df = 3$, $p = 7.6 \times 10^{-4}$) and response detection frequency ($N = 251$, $w = 16.35$, $df = 3$, $p = 9.6 \times 10^{-4}$) correlated positively with discharge outcome. Using post hoc tests, response amplitude and response frequency were associated with recovery of consciousness at discharge and 6 months (Fig. 5, Supplementary Fig. 4).

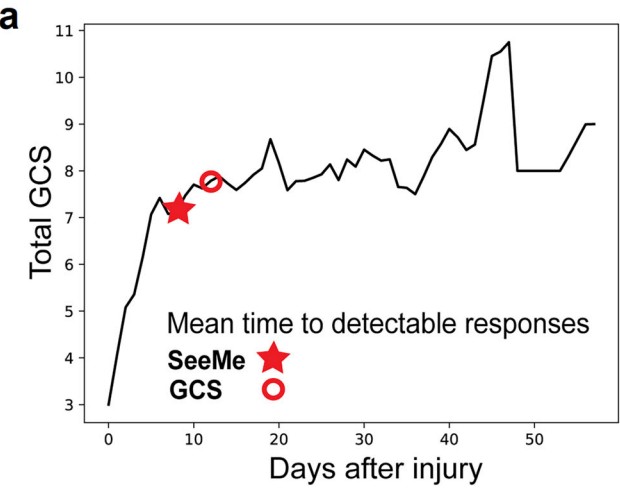

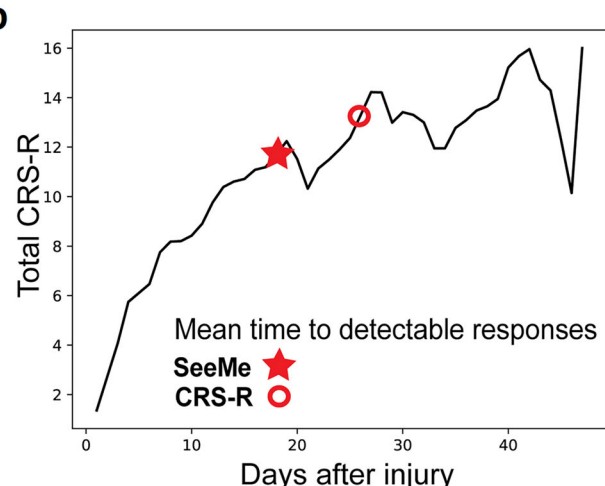

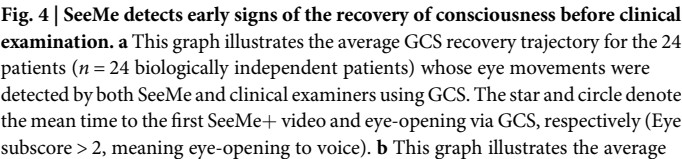

**Fig. 4 | SeeMe detects early signs of the recovery of consciousness before clinical examination. a** This graph illustrates the average GCS recovery trajectory for the 24 patients (*n* = 24 biologically independent patients) whose eye movements were detected by both SeeMe and clinical examiners using GCS. The star and circle denote the mean time to the first SeeMe+ video and eye-opening via GCS, respectively (Eye subscore > 2, meaning eye-opening to voice). **b** This graph illustrates the average CRS-R recovery trajectory of the seven patients (*n* = 7 biologically independent patients) whose mouth movements were detected by both SeeMe and clinical examiners using CRS-R. The star and circle denote the mean time to the first SeeMe + video and mouth response via CRS-R, respectively. (Auditory subscore > 2, meaning command following).

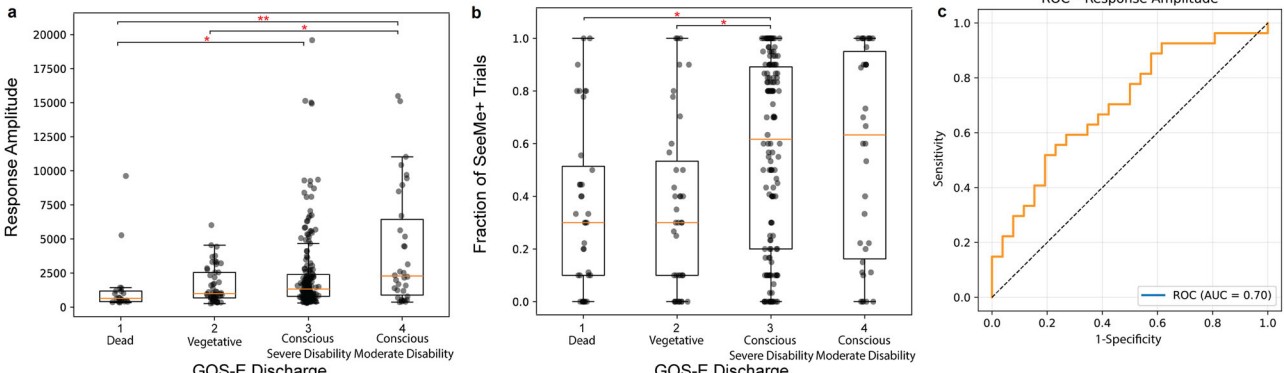

**Fig. 5 | SeeMe responses correlate with patients' outcomes.** Box plots comparing the average maximum amplitude (**a**) per SeeMe+ video and the average fraction of SeeMe+ trials (**b**) per video for all 36 ABI patients (*n* = 36 biologically independent patients), based on the patient's GOS-E at discharge. The patients are divided into dead (GOS-E = 1), vegetative (GOS-E = 2), conscious severe disability (GOS-E = 3), and conscious moderate disability (GOS-E = 4). The yellow line represents the median of each data set. Kruskal–Wallis test followed by post-hoc pairwise comparisons using the Dunn–Bonferroni approach; *$p < 0.05$, **$p < 0.01$. **c** AUC-ROC curve illustrating classification performance based on amplitude features from the final day of SeeMe recording (AUC = 0.70, on average 15.3 days after injury, while the discharge was 42.6 days after injury).

## Validating SeeMe results using secondary classifiers

To address the concern that movements may not reflect command following, we tested whether we could identify which command was presented based on the facial movements. For example, the facial movements in response to the Open your eyes command should be easily distinguished from movements in response to the Show me a smile command. We thus trained and tested a deep neural network classifier to predict which command was presented, given the facial movement from SeeMe+ trials. We achieved 81, 37, and 47% accuracy for eye, tongue, and smile commands, respectively, with 65% overall accuracy, i.e., we could tell what command had been presented based on the facial movements alone in most trials (Supplementary Fig. 5).

## Discussion

In this study, we developed a high-resolution computer-vision-based tool, SeeMe, to objectively assess the recovery of stimulus-evoked movements after ABI. SeeMe demonstrates the existence of low-amplitude stimulus-

evoked movements in recovering ABI patients that are undetectable by the naked eye (Fig. 2). These micromovements were identified by SeeMe in more patients than clinical examination days to weeks before the manifestation of overt signs of consciousness (Fig. 4). This lead time could be critical for patient care and outcomes. Furthermore, the amplitude and frequency of these detected movements significantly correlate with the outcome of these patients at discharge, as measured by the GOS-E score (Fig. 5). Finally, our deep neural network classifier showed reasonable performance in determining the command presented using facial movements alone, suggesting the specificity of the detected responses (Supplementary Fig. 5).

Our findings are critical in light of recent studies that demonstrate a significant fraction of ABI patients who seem unresponsive can be covertly aware[1,2], with potential for recovery. Recent efforts to detect early signs of covert consciousness have leveraged diverse neuroimaging and electrophysiological modalities, each with its respective strengths and weaknesses. In a landmark study that was the culmination of decades of research, Bodien

and colleagues employed EEG and fMRI during mental imagery tasks and detected covert responses in nearly 25% of patients with disorders of consciousness, although there was also a notably high false negative rate of 62% in subjects who did have observable responses to commands[2]. Similarly, Schnakers et al. demonstrated the utility of cognitive evoked-related potentials during active behavioral tasks in detecting capacity for volitional activity in minimally conscious patients, though there was again reduced sensitivity, possibly due to logistical limitations[25]. Casarotto and colleagues demonstrated near-perfect sensitivity and specificity in detecting covert consciousness by utilizing the Perturbation Complexity Index (a robust measure of the capacity of thalamocortical networks to integrate information, which can be captured by EEG responses to transcranial magnetic stimulation) in a cohort with independently verified conscious and unconscious states[26]. In an externally validated cohort, this metric identified covert consciousness in 9 of 43 (20.9%) unresponsive patients. Other studies utilizing high-density EEG and advanced machine learning approaches have also demonstrated potential in detecting covert consciousness; however, their translation to routine clinical practice is limited by the need for extensive computational resources and technical expertise[27,28]. In contrast, SeeMe offers a readily interpretable bedside resource that can be used to detect subtle volitional facial movements. While formal sensitivity and specificity metrics are pending validation, our results suggest that SeeMe may offer another alternative for covert consciousness detection. Future work will synergize SeeMe with other current approaches, particularly EEG, to improve early detection of covert consciousness.

It is worth noting that the absence of early detection should never be interpreted as the absence of potential. Meaningful recovery can occur well after prolonged in-hospital unconsciousness, and such potential patients should not be categorically excluded from receiving rehabilitative care. Rather, the early detection of consciousness can help facilitate earlier initiation of intensive rehabilitation strategies, which is associated with improved outcomes[29,30]. In addition, it may provide an opportunity to open a channel of communication with these patients. SeeMe is intended to complement, not replace, longitudinal care and observation. Further work should be performed to determine the prognostic utility of positive and negative SeeMe results and validate these findings using more longitudinal, long-term data following discharge. Particular attention should focus on initially negative SeeMe results for patients who subsequently recover consciousness.

Several limitations should be mentioned. While we believe our technique reliably identifies movement to auditory command, there are a variety of reasons why a patient might not respond to the computer while responding to a clinical examiner (i.e., as in the three patients who opened eyes to voice before SeeMe). These patients may have had fluctuations in arousal or even willingness to participate. Experienced clinicians note notable variability in neurological examination in all patients. Thus, it is not necessarily indicative of any problems that SeeMe detected movements after clinicians in some cases. This is an exceedingly challenging population to study, and we eagerly await the availability of life support techniques that do not require as much sedation to maintain ventilation. Conversely, SeeMe identified responses in five patients who did not demonstrate command-following on clinical examination at any point during the study period. Remarkably, three of the five patients recovered consciousness before hospital discharge, but after study procedures. The remaining two patients did not recover during the hospitalization. While the possibility of false positives for these two patients cannot be excluded, it is well recognized that some patients regain consciousness only after discharge. Thus, SeeMe should be used with other clinical studies and assessments to comprehensively inform prognosis and guide treatment planning. To validate our findings further, future studies should incorporate objective measures of motor activity, such as electromyography, and investigate the prognostic potential of SeeMe further. A second limitation of our study is that the subjects had different types of injuries, which could affect their ability to respond to auditory stimulation commands when awake.

However, given that our technique measures stimulus-evoked behavior, it seems likely that the ability to measure movement is generalizable across disease processes. In addition, on average, our population has high response rates to auditory input (87%). Similar studies in ABI have similarly used mixed cohorts[1,31]. Injuries involving muscle groups of the ROIs explored in this study could have also substantially limited responses to commands. Thus, future work should explore responses to other commands (i.e., close your hands, wiggle your toes). A third limitation is the analyzability of mouth videos. Ventilation equipment often blocked the ROI for the mouth and lower facial region in these critically ill patients. However, 16/17 had detectable responses to the commands in patients with analyzable video. Fourth, our recording and examination sessions were limited due to the nature of conducting research with the vulnerable population of critically ill patients. Sessions were occasionally prevented by clinical instability or family preference, and medications, such as sedatives and muscle paralytics, impede the detection of motor movements. Fifth, stimuli-evoked behavior can be influenced by common complications that arise in the care of critically ill patients. Our analysis of SeeMe+ versus SeeMe- videos obtained within one week after well-known in-hospital complications[23] suggests that patients who develop hospital-acquired pneumonia or pulmonary embolism are less likely to have SeeMe+ detected movements in the week following either complication (Supplementary Table 1), this is due to an increased level of sedation, to manage critical clinical conditions such as intubation, reduction of metabolic demand in the brain, or other therapeutic interventions associated with these complications[32–35]. Conversely, patients exhibiting appreciably increased movements, as seen in delirium, appear to have more SeeMe+ videos up to a week from delirium onset. This can be explained by increased arousal in these patients, as we previously reported with agitation[36]. These findings suggest that SeeMe may be able to distinguish command following in patients suffering from delirium. Although SeeMe's original application was intended for the objective assessment of the recovery of consciousness after ABI, in future work, we plan to assess the capacity for following commands in patients suffering from delirium.

## Conclusion

Due to growing recognition that a substantial portion of ABI patients who are considered unresponsive are actually covertly conscious and capable of cognitive function, there is a pressing need for tools that can detect early signs of consciousness more reliably. Clinicians have an ethical imperative to identify patients who are capable of demonstrating signs of awareness and develop treatment strategies accordingly. Thus, SeeMe is a technology that can be used in conjunction with other clinical modalities to provide crucial and valuable information needed to facilitate important discussions between clinicians and family members about their loved one's care.

Ultimately, this tool opens the door for quantitative research targeting the development of novel, as far as we are aware, approaches to facilitate communication and/or recovery of consciousness in these patients. Patients with low-amplitude movements represent an appealing population for brain-computer interfaces to enable communication because these patients are conscious and able to interact with the environment[37]. Moreover, detecting and coupling low-amplitude movements with electrical stimulation has been an effective therapeutic strategy in stroke, so a similar approach may be practical in ABI patients[38,39]. Future therapeutic closed-loop stimulation systems to enhance consciousness will require a real-time objective measure of consciousness, as we have developed[40]. SeeMe, our first-in-class objective measure of consciousness in ABI patients, has immediate use in ABI patient assessment and monitoring.

## Data availability

The source data underlying Table 1, Fig. 4a, b, Supplementary Fig. 4a, b, and Supplementary Table 1 are provided as Supplementary Data 4. The remaining data that support the findings of this study are available on request from the corresponding author. Video data are not publicly available due to privacy or ethical restrictions.

## Code availability

The authors will make the SeeMe code available to other researchers upon reasonable request. Interested researchers may contact the corresponding author, S.M., for further details.

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

## Acknowledgements

This work was supported by OVPR SEED Funding, SUNY SEED, and Targeted Research Opportunity Program (Stony Brook TRO, Fusion). We thank Raphael Davis, the Department of Neurosurgery, and the Paige Elizabeth Keely Foundation for their ongoing support, Brandon Foreman, Nicholas Schiff, Len Polizzotto, and Karin Diserens for helpful discussions of this manuscript, and the patients and their families for participating in this research.

## Author contributions

X.C., C.M. and S.M. contributed to the conception and design of the study. X.C., S.S., J.R., N.C., J.S., C.U., Y.H., S.M.M.A., C.W., R.K., X.Z., A.F., J.S.,

K.B., C.C., J.D., P.D., C.M. and S.M. contributed to the acquisition and analysis of the data. X.C., S.S., J.R., N.C., J.S., C.U., Y.H., S.M.M.A., P.D., C.M. and S.M. contributed to the drafting of the manuscript and/or figures.

## Competing interests

The authors declare the following competing interests: S.M., C.B.M., and J.D. disclose interests in SeeMe Technologies, a startup based on the methodology described in this report.
