## [Transparent Peer Review file · Communications Medicine]

Computer Vision Detects Covert Voluntary Facial Movements in Unresponsive Brain Injury Patients

Corresponding Author: Dr Sima Mofakham

Version 0:

Reviewer comments:

Reviewer #1

(Remarks to the Author)

In this manuscript, Cheng and colleagues present SeeMe+, a new method for the detection of covert voluntary facial movements applied to the early detection of consciousness in brain injured patients.

This is a valuable work that makes a significant step forward in the open issue of detecting covert consciousness. The method proposed by the authors builds on traditional approaches for detecting consciousness, but leverages computer vision to enhance their diagnostic and prognostic power. Notably, this approach has the nontrivial potential to be quite straightforwardly implemented in the clinical practice. The paper has the potential to impact the field. The study is clearly presented and written in all its parts, and it is carefully conducted from an experimental, analytical, and statistical viewpoint. My only notable criticism is that the low accuracy in detecting mouth movements as specific of the mouth area (close to chance level) should be clearly presented to the reader and discussed. Also, a more exhaustive discussion of the current methods to detect covert consciousness would allow the reader to better position the manuscript in the current scientific landscape. Additionally, sharing the SeeMe code would enhance the impact and reproducibility of the work.

1) Example cases can be further detailed, for example reporting the lobes lesioned by the traumatic event and similar clinical details.

2) The authors report, in the supplementary results, that 5 patients were SeeMe+ but Clinical Exam-. This important result is worth being discussed.

3) The authors report 81% accuracy for eye commands and 65% overall accuracy. For sake of clarity, I recommend to report also the accuracy for tongue command and smile command. Since eFigure 5 suggest that these two accuracies are marginally above chance level, I suggest to discuss this finding, e.g. can these be considered non-specific startle reactions? Or the low accuracy may be due to the inclusion of two distinct movements of the mouth area that may be hard to separate?

5) To enhance the reproducibility and impact of the study, I suggest the authors to share the code used for SeeMe and subsequent analyses.

6) I suggest the authors to discuss their method in comparison to other recent approaches for the early detection of covert consciousness, such as Casarotto, *Annals of Neurology*, 2016; Franzova, *Brain*, 2023; Schnakers, *Neurology*, 2008; Sitt, *Brain*, 2014; Bodien, *New England Journal of Medicine*, 2024; for a review Comanducci, *Clinical Neurophysiology*, 2020. For example, to provide the readers with a broader perspective, it would be useful to compare (when possible) the proposed method in terms of timing, sensitivity, and specificity to these other approaches, even just at a speculative level.

7) Optionally, the impact of the current manuscript could increase by providing few additional steps towards the clinical implementation of this approach. For example, comparing the AUC-ROC analysis between conscious (GOS-E=1/2) and unconscious patients (GOS-E=3/4) would be a valuable addition to Figure 5. In the same spirit, which threshold would maximize the separation between conscious/unconscious outcomes in terms of response amplitude and fraction of SeeMe+ trials?

Minors:

Typo: two points in line 352 after "i.e. "

Typo: two points in supplementary materials line 296, after "recovery"

Note that the legends of eTable 2 and 3 are hard to read because they overlap with the line numeration.

The authors report the distance from between the camera and the patient's face in feet. I recommend reporting it in international system units (meters).

Simone Russo

Reviewer #2

(Remarks to the Author)

This is an interesting manuscript addressing the assessment of consciousness in patients with acute brain injury. It is now well known that patients with "cognitive motor dissociation" may have conscious awareness without showing overt movement to command. The authors of this study explore the question of whether "subthreshold" movements to command may precede overt movement and, therefore, whether methods with increased movement sensitivity might be useful in clinical assessment.

The study is well designed and conducted. Statistical analyses are simple but clear. Its results can be considered from 2 perspectives: the overall concept being addressed; and the specific assessment technology being applied.

With regard to the former, the authors have convincingly demonstrated that subtle command-related movements emerge before more obvious responses to command. They show that their detection system identifies signs of consciousness earlier and identifies more trials in which the motor behavior coincided with the requested response. This suggests that cognitive motor dissociation is not all or nothing, but that patients may have low amplitude motor responses as well. Accordingly, the data also suggest that any method that increases the sensitivity to detection of motor outflow could augment assessment results.

The study's implications for practical assessment are less clear. First is the question of whether this system is the optimal system for balancing sensitivity with specificity. Candidates for increasing sensitivity include not only existing functional imaging assessments or covert command following, but also EMG measures of sub-movement motor discharges and subtle motion capture, as in this study. And of course, different requested movements may have different probabilities of revealing consciousness. Thus, further work would be needed to identify the system with the greatest diagnostic accuracy. A larger question is what hinges on detecting consciousness a few days earlier or with a bit greater confidence? In the current US climate in which treatment is often withdrawn early in the absence of signs of consciousness, such evidence may have a major influence on care decisions. But in recently updated guidelines, it has been noted that good recovery is possible even for patients who spend considerable intervals unconscious and, therefore, care plans should allow more time for recovery rather than attempting to sort people extremely early by prognosis. Thus, it would be useful to discuss how such an assessment technique is likely to be used and how such a technique should be used.

Version 1:

Reviewer comments:

Reviewer #1

(Remarks to the Author)

The authors addressed all my concerns and significantly improved the quality of the manuscript.

Simone Russo

Reviewer #2

(Remarks to the Author)

The authors have nicely addressed the technical comments about the manuscript, and have discussed the likely differences in sensitivity/specificity among different candidate behaviors and potential reasons for these differences. However, I still find their discussion of the ethical implications of the assessment confusing and contradictory.

As noted in my prior review, since relatively prolonged unconsciousness is still compatible with a favorable recovery profile, we need a clearer explanation of the benefit of detecting consciousness a few days earlier. As requested, the authors have added to the discussion the fact that recovery can still occur after prolonged unconsciousness, but go on to say (pg. 17), "A

key goal of ABI care is the identification of patients with rehabilitative potential. An early return to consciousness predicts superior outcomes and identifies a subset of patients on whom scarce resources for rehabilitative care can be focused.” On page 20, they state that, “Due to growing recognition that meaningful recovery can occur even after prolonged unconsciousness, there is a pressing need for tools that can detect early signs of consciousness more reliably.” These are confusing and contradictory statements. If patients with prolonged unconsciousness still have recovery potential, then even some of the patients who test negative with SeeMe could likely still have good recoveries. It’s impossible to identify a group of patients on whom to “focus scarce resources” without simultaneously identifying the rest of the group from whom those resources may be denied. One might equally well argue that the “pressing need” is to provide high quality care for long enough to allow behavioral recovery, which obviates the need to predict very early which patients will experience that recovery. This is the predominant care model in the Netherlands and the UK, which rely much less on early high-tech exploration of consciousness, and much more on longitudinal care and observation. In other words, the focus on early detection of consciousness is linked to the health care system’s desire to limit post-acute care. Thus, if the authors wish to argue that their measure can be useful in making ethically important decisions such as access to rehabilitation services, they need to report its accuracy with respect to long-term outcomes – in particular what proportion of patients with negative SeeMe results in the hospital go on to have significant recovery. Otherwise, it’s possible that its use simply increases the confidence with which clinicians inappropriately deny care. Or they could argue that SeeMe might have the potential to provide an accurate early prognosis and that question needs further study (though this argument isn’t terribly plausible early predictors, in general, have more ‘noise’ than later ones). Alternatively, they could shift their focus from “early detection” to detection at all. At whatever point consciousness is detected, it shifts the focus of treatment from a passive focus to an active one, which is a clinically and ethically meaningful shift.

Version 2:

Reviewer comments:

Reviewer #2

(Remarks to the Author)

I think the authors' recent modifications adequately address my ethical concerns.

**Neurosciences Institute
Department of Neurological Surgery
NY Spine & Brain Surgery, U.F.P.C.**

Dear Reviewers,

Thank you for considering our manuscript and for your constructive comments. We believe addressing these comments improved our manuscript. Please see our responses to each reviewer's suggestions below:

Reviewer #1:

1) Example cases can be further detailed, for example reporting the lobes lesioned by the traumatic event and similar clinical details.

Answer: We have added further radiographic descriptions of the brain regions affected by the traumatic events and details on other clinical injuries.

Subject 13:

“Initial CTH showed subarachnoid hemorrhage involving the left sylvian fissure and adjacent sulci, epidural hemorrhage around the right anterior temporal lobe, numerous skull and facial fractures, and pneumocephalus. Other injuries on presentation were right hemopneumothorax, rib fractures, dissection of the descending thoracic aorta, and left acetabular fractures.”

Subject 2: “Initial CTH showed left subdural hematoma along the frontal and parietal convexity with effacement of adjacent sulci; remaining workup showed bilateral lower extremity injuries.”

2) The authors report, in the supplementary results, that 5 patients were SeeMe+ but Clinical Exam-. This important result is worth being discussed.

Answer: Thank you for raising this excellent point. As stated in our supplementary section, 5 patients determined to be SeeMe+ were not Clinical Exam+ during the study period. Remarkably, three of the five patients recovered consciousness at some point before hospital discharge. The remaining two patients did not recover during the hospitalization.

We have added the following section in the discussion to expand upon our findings:

“Conversely, SeeMe identified responses in five patients who did not demonstrate command-following on clinical examination at any point during the study period. Remarkably, three of the five patients recovered consciousness before hospital discharge, but after study procedures. The remaining two patients did not recover during the hospitalization. While the possibility of false positives for these two patients cannot be excluded, it is well recognized that some patients regain consciousness only after discharge. Thus, SeeMe should be used with other clinical studies and assessments to comprehensively inform prognosis and guide treatment planning. To validate our findings further, future studies should incorporate objective measures of motor activity, such as electromyography (EMG), and investigate the prognostic potential of SeeMe further.”

3) The authors report 81% accuracy for eye commands and 65% overall accuracy. For sake of clarity, I recommend to report also the accuracy for tongue command and smile command. Since eFigure 5 suggest that these two accuracies are marginally above chance level, I suggest to discuss this finding, e.g. can these be considered non-specific startle reactions? Or the low accuracy may be due to the inclusion of two distinct movements of the mouth area that may be hard to separate?

Answer: We have now also reported the accuracy for tongue and smile commands (37% and 47%, respectively, while the chance level is 33%) in the discussion section and Supplementary Figure 5. There are a number of possibilities for the relatively lower accuracy of the mouth commands compared to the eye commands.

- 1.) As the reviewer noted, distinguishing between the two mouth commands may be challenging, especially as we utilize similar regions of interest for analysis (see figure below; the participant permitted to share the photograph). Similarly, the complexity and variety of movements are much higher for mouth commands than eye commands and can thus be challenging to model.
- 2.) Each of the selected commands was chosen to elicit movements that engage different muscle groups. There may be differences in the level of effort required to produce a movement (i.e. sticking out tongue may require more effort than opening the eyes). In addition, often eye movement accompanied the mouth movement, which reduced the decoder accuracy.
- 3.) Additionally, fewer analyzable videos were available for mouth commands, because we could only use those in which the patients had already been extubated. Often in these patients, prolonged immobilization in general can result in muscular atrophy and weakness, and intubation is no exception. Thus, patients may reasonably demonstrate oral motor weakness following extubation that is independent of impairment of consciousness. Our team is working

on future projects that will allow us to capture mouth movements while the patient is intubated and explore the utility of other commands (i.e., close your hands).

5) To enhance the reproducibility and impact of the study, I suggest the authors to share the code used for SeeMe and subsequent analyses.

Answer: We agree with the reviewers, and the authors will make the SeeMe code available to other researchers upon reasonable request. This tool has the potential to significantly transform the way neurological examinations are conducted in patients with disorders of consciousness. We aim to develop a widely accessible application shortly and share it with the broader medical and scientific community.

6) I suggest the authors to discuss their method in comparison to other recent approaches for the early detection of covert consciousness, such as Casarotto, *Annals of Neurology*, 2016; Franzova, *Brain*, 2023; Schnakers, *Neurology*, 2008; Sitt, *Brain*, 2014; Bodien, *New England Journal of Medicine*, 2024; for a review Comanducci, *Clinical Neurophysiology*, 2020. For example, to provide the readers with a broader perspective, it would be useful to compare (when possible) the proposed method in terms of timing, sensitivity, and specificity to these other approaches, even just at a speculative level.

Answer: We appreciate the reviewer’s suggestion. We have added the following to the discussion section:

“Recent efforts to detect early signs of covert consciousness have leveraged diverse neuroimaging and electrophysiological modalities, each with their respective strengths and weaknesses. In a landmark study that was the culmination of decades of research, Bodien and colleagues employed EEG and fMRI during mental imagery tasks and detected covert responses

in nearly 25% of patients with disorders of consciousness although there was also a notably high false negative rate of 62% in subjects who did have observable responses to commands.¹ Similarly, Schnakers et al. demonstrated the utility of cognitive evoked-related potentials during active behavioral tasks in detecting capacity for volitional activity in minimally conscious patients though there was again reduced sensitivity possibly due to logistical limitations.² Casarotto and colleagues demonstrated near perfect sensitivity and specificity in detecting covert consciousness by utilizing the Perturbation Complexity Index (a robust measure of the capacity of thalamocortical networks to integrate information, which can be captured by EEG responses to transcranial magnetic stimulation) in a cohort with independently verified conscious and unconscious states.³ In an externally validated cohort, this metric identified covert consciousness in 9 of 43 (20.9%) unresponsive patients. Other studies utilizing high-density EEG and advanced machine learning approaches have also demonstrated potential in detecting covert consciousness; however, their translation to routine clinical practice is limited by the need for extensive computational resources and technical expertise.^{4,5} In contrast, SeeMe offers a readily interpretable bedside resource that can be used to detect subtle volitional facial movements. While formal sensitivity and specificity metrics are pending validation, our results suggest that SeeMe may offer another alternative for covert consciousness detection. Future work will synergize SeeMe with other current approaches, particularly EEG, to improve early detection of covert consciousness.”

1. Bodien, Y. G. *et al.* Cognitive motor dissociation in disorders of consciousness. *N. Engl. J. Med.* **391**, 598–608 (2024).
2. Schnakers, C. *et al.* Voluntary brain processing in disorders of consciousness. *Neurology* **71**, 1614–1620 (2008).
3. Casarotto, S. *et al.* Stratification of unresponsive patients by an independently validated index of brain complexity: Complexity Index. *Ann. Neurol.* **80**, 718–729 (2016).
4. Franzova, E. *et al.* Injury patterns associated with cognitive motor dissociation. *Brain* **146**, 4645–4658 (2023).
5. Sitt, J. D. *et al.* Large scale screening of neural signatures of consciousness in patients in a vegetative or minimally conscious state. *Brain* **137**, 2258–2270 (2014).

7) Optionally, the impact of the current manuscript could increase by providing a few additional steps towards the clinical implementation of this approach. For example, comparing the AUC-ROC analysis between conscious (GOS-E=1/2) and unconscious patients (GOS-E=3/4) would be a valuable addition to Figure 5. In the same spirit, which threshold would maximize the separation between conscious/unconscious outcomes in terms of response amplitude and fraction of SeeMe+ trials?

Answer: This is an excellent recommendation. The AUC-ROC results for amplitude are shown in the figures below. We plotted the AUC-ROC results for amplitude based on the final day of SeeMe recording, with an AUC of **0.70**. We added this figure to **Figure 5**. Our results suggest a correlation between SeeMe+ response amplitude/frequency and conscious outcome as measured by GOS-E. SeeMe assessments took place during the acute stage of brain injury to capture any emerging command-following behavior. However, many of these patients experienced extended hospital stays during which several factors may influence their consciousness level, including medical complications and fluctuations in consciousness. Thus, in our future work, we plan to explore the prognostic value of SeeMe in this population.

Minors:

Typo: two points in line 352 after "i.e. "

Typo: two points in supplementary materials line 296, after "recovery"

Note that the legends of eTable 2 and 3 are hard to read because they overlap with the line numeration.

The authors report the distance from between the camera and the patient's face in feet. I recommend reporting it in international system units (meters).

Answer: Thank you for your helpful comments. We have addressed these points and made the corresponding corrections in the manuscript and supplementary materials.

Reviewer #2 (Remarks to the Author):

1) The study's implications for practical assessment are less clear. First is the question of whether this system is the optimal system for balancing sensitivity with specificity.

Candidates for increasing sensitivity include not only existing functional imaging assessments or covert command following, but also EMG measures of sub-movement motor discharges and subtle motion capture, as in this study. And of course, different requested movements may have different probabilities of revealing consciousness.

Answer: We thank the reviewer for their thoughtful and insightful comment. We added a subpanel to **Figure 5** showing the sensitivity/specificity of SeeMe outcome prediction based on response amplitudes. Our data (**Figure 4**) shows that recovery occurs **4-8 days earlier** than what is captured by standard clinical examination, such as the Glasgow Coma Scale (GCS) or Coma Recovery Scale - Revised (CRS-R). As the reviewer has noted, future validation using more precise modalities, such as functional imaging (fMRI, EEG) and motor activity monitoring (EMG), will be essential to further establish the utility of SeeMe. We also agree that expanding the range of commands and motor responses (e.g. close your hands, wiggle your toes) should be assessed to determine their potential for detecting early recovery of consciousness. We recognize these as critical future directions and are actively pursuing efforts to incorporate them into our ongoing and forthcoming research. We have added this to our discussion.

2) Thus, further work would be needed to identify the system with the greatest diagnostic accuracy. A larger question is what hinges on detecting consciousness a few days earlier or with a bit greater confidence? In the current US climate in which treatment is often withdrawn early in the absence of signs of consciousness, such evidence may have a major influence on care decisions. But in recently updated guidelines, it has been noted that good recovery is possible even for patients who spend considerable intervals unconscious and, therefore, care plans should allow more time for recovery rather than attempting to sort people extremely early by prognosis. Thus, it would be useful to discuss how such an assessment technique is likely to be used and how such a technique should be used.

Answer: This is an excellent point. Our goal is not to promote early sorting for prognostic purposes but to empower clinicians and families with more accurate, behaviorally grounded information to guide care planning and avoid premature withdrawal of life-sustaining treatment. Regarding the aforementioned guidelines, SeeMe enables earlier detection of motor commands that clinicians could not detect using standard clinical examinations. Thus, it may facilitate discussions between family members and physicians regarding their loved one's care. Ultimately, SeeMe adds another layer of information about the functional state of the brain and capacity for command following. However, SeeMe should be used in the bigger picture context, along with other clinical modalities, to provide families and clinicians with accurate and valuable information needed to make important decisions on their loved one's behalf. We have added this to our discussion.

Sincerely yours,

Sima Mofakham, Ph.D.

Director of Research and Assistant Professor

Department of Neurological Surgery

Stony Brook University

101 Nicolls Road

Stony Brook, NY 11794

sima.mofakham@stonybrookmedicine.edu

**Neurosciences Institute
Department of Neurological Surgery
NY Spine & Brain Surgery, U.F.P.C.**

Dear Reviewers,

Thank you for considering our manuscript and for your constructive comments. We believe addressing these comments improved our manuscript. Please see our responses to each reviewer's suggestions below:

Reviewer #1:

1) The authors addressed all my concerns and significantly improved the quality of the manuscript.

Answer: Thank you so much for your previous comments. They significantly improved our manuscript.

Reviewer #2:

1) The authors have nicely addressed the technical comments about the manuscript, and have discussed the likely differences in sensitivity/specificity among different candidate behaviors and potential reasons for these differences.

Answer: Thank you for your response!

2) However, I still find their discussion of the ethical implications of the assessment confusing and contradictory. As noted in my prior review, since relatively prolonged unconsciousness is still compatible with a favorable recovery profile, we need a clearer explanation of the benefit of detecting consciousness a few days earlier. As requested, the authors have added to the discussion the fact that recovery can still occur after prolonged unconsciousness, but go on to say (pg. 17), "A key goal of ABI care is the identification of patients with rehabilitative potential. An early return to consciousness predicts superior outcomes and identifies a subset of patients on whom scarce resources for rehabilitative care can be focused." On page 20, they state that, "Due to growing recognition that meaningful recovery can occur even after prolonged unconsciousness, there is a pressing need for tools that can detect early signs of consciousness more reliably."

These are confusing and contradictory statements. If patients with prolonged unconsciousness still have recovery potential, then even some of the patients who test negative with SeeMe could likely still have good recoveries. It's impossible to identify a group of patients on whom to "focus scarce resources" without simultaneously identifying the rest of the group from whom those resources may be denied.

Answer: We agree with the reviewer that the ethical implications of SeeMe should be addressed clearly. In our manuscript, we acknowledge that meaningful recovery can occur even after prolonged unconsciousness that persists through discharge. *Patients who do not exhibit early signs of consciousness should not be categorically excluded from rehabilitative care.* We have revised our discussion to emphasize that "absence of early detection of recovery should never be interpreted as absence of potential." The SeeMe system *is not designed to rule out recovery* but to *improve early identification of patients who may benefit from early therapeutic engagement and communication.* In our future work, we are trying to use SeeMe to provide a way to communicate with these covertly conscious patients.

We have reworded sections (pages 17 and 19) to clarify that our primary ethical argument is **NOT** about **reallocating or denying care** based on early detection. Rather, we argue that detecting consciousness earlier - when it is present - **can help initiate rehabilitative interventions sooner, which is associated with improved outcomes** (Whyte et al., 2013; Nakase-Richardson et al., 2020). We maintain that SeeMe can **reduce the risk of inappropriate withdrawal of care** in patients with covert consciousness as there is currently no easy way to identify these patients. Importantly, this approach should **supplement rather than replace** longitudinal care and observation. SeeMe is like other medical devices such as EEG and MRI that can provide valuable information about the functional state of the brain when used in conjunction with other modalities (EEG, MRI). The benefit of detecting consciousness earlier lies in improving diagnostic precision during a highly ambiguous and emotionally charged period of care. In current practice, the absence of overt behavioral signs often leads to misclassification of conscious patients as unconscious, resulting in missed opportunities for early engagement, communication attempts, or basic interventions like pain management and structured stimulation.

We have underscored that **a negative result from SeeMe should never be used to deny care.** We acknowledge the risk, noted by the reviewer, that new technologies could inadvertently reinforce premature withdrawal of care if not properly contextualized. We now more explicitly state that the SeeMe platform must be validated longitudinally to determine how its outputs relate to long-term outcomes, and that its deployment must be accompanied by rigorous clinical training and ethical oversight. We agree that for SeeMe to support ethically meaningful decisions regarding resource allocation or prognosis, its predictive validity with respect to long-term outcomes must be demonstrated. Our work suggests that SeeMe correlates positively with

outcomes as measured by GOS-E at 6-months post-discharge (Supplementary Figure 4). We plan to conduct future studies, intended to examine SeeMe performance with more longitudinal data, including cases in which early assessments are negative but recovery later occurs.

3) One might equally well argue that the “pressing need” is to provide high quality care for long enough to allow behavioral recovery, which obviates the need to predict very early which patients will experience that recovery. This is the predominant care model in the Netherlands and the UK, which rely much less on early high-tech exploration of consciousness, and much more on longitudinal care and observation. In other words, the focus on early detection of consciousness is linked to the health care system’s desire to limit post-acute care.

Answer: We appreciate this perspective and agree that longitudinal care is essential. SeeMe is **NOT** intended to justify early withdrawal of care or serve system-level cost priorities. Negative SeeMe results do not mean the lack of potential for recovery. SeeMe is meant to help patients by identifying covert consciousness, that might otherwise go undetected, to adjust treatment and increased efforts for communication with these patients. Its value lies in expanding, not limiting, clinical insight and guiding ethically sound decision-making.

4) Thus, if the authors wish to argue that their measure can be useful in making ethically important decisions such as access to rehabilitation services, they need to report its accuracy with respect to long-term outcomes – in particular what proportion of patients with negative SeeMe results in the hospital go on to have significant recovery. Otherwise, it’s possible that its use simply increases the confidence with which clinicians inappropriately deny care.

Answer: We agree that all patients should have access to rehabilitation, regardless of early prognostic indicators. However, in practice, decisions to limit care are sometimes made prematurely. In such cases, SeeMe may help preserve access to care by revealing signs of consciousness that might otherwise be missed, offering a chance for continued treatment and recovery.

5) Or they could argue that SeeMe might have the potential to provide an accurate early prognosis and that question needs further study (though this argument isn’t terribly plausible early predictors, in general, have more ‘noise’ than later ones). Alternatively, they could shift their focus from “early detection” to detection at all. At whatever point consciousness is detected, it shifts the focus of treatment from a passive focus to an active one, which is a clinically and ethically meaningful shift.

Answer: We appreciate the reviewer’s concern and agree that the ethical implications of SeeMe require clear articulation. We have revised our discussion (pages 17 and 19) to clarify that early detection is not intended as a triage tool to limit care, but rather as an opportunity to engage patients earlier when signs of consciousness are present. We fully acknowledge that meaningful recovery can occur even after prolonged unconsciousness, and emphasize that a negative SeeMe result should never be interpreted as absence of potential. SeeMe is designed to supplement, not

replace, longitudinal observation and multimodal assessment (e.g., EEG, MRI). It provides an additional layer of information to support clinicians during a period often marked by diagnostic uncertainty. Our revised text underscores that SeeMe is not a tool for deciding who “deserves” care or when care should be withdrawn. On the contrary, it may help prevent premature decisions by identifying covert consciousness that would otherwise go unrecognized. We have added that SeeMe’s clinical use must be accompanied by further validation of its prognostic value.

Sincerely yours,

Sima Mofakham, Ph.D.

Director of Research and Assistant Professor

Department of Neurological Surgery

Stony Brook University

101 Nicolls Road

Stony Brook, NY 11794

sima.mofakham@stonybrookmedicine.edu